# Applications of Nanomaterials for Heavy Metal Removal from Water and Soil: A Review

**Guo Yu** [1,2], **Xinshuai Wang** [1], **Jie Liu** [1,2,3,*], **Pingping Jiang** [1,2,*], **Shaohong You** [1,2,3], **Na Ding** [1], **Qianjun Guo** [1] and **Fanyu Lin** [4]

[1] College of Environmental Science and Engineering, Guilin University of Technology, Guilin 541004, China; yuguo@glut.edu.cn (G.Y.); w_xinshuai@163.com (X.W.); youshaohong@glut.edu.cn (S.Y.); nading001@163.com (N.D.); tracy_gqj@163.com (Q.G.)

[2] Guangxi Key Laboratory of Environmental Pollution Control Theory and Technology for Science and Education Combined with Science and Technology Innovation Base, Guilin University of Technology, Guilin 541004, China

[3] Technical Innovation Center of Mine Geological Environmental Restoration Engineering in Southern Karst Area, Ministry of Natural Resources, Guilin 541004, China

[4] Third Institute of Oceanography, Ministry of Natural Resources, Xiamen 200136, China; linfanyu@tio.org.cn

[*] Correspondence: liujie@glut.edu.cn (J.L.); jiangpp@glut.edu.cn (P.J.)

**Abstract:** Heavy metals are toxic and non-biodegradable environmental contaminants that seriously threaten human health. The remediation of heavy metal-contaminated water and soil is an urgent issue from both environmental and biological points of view. Recently, nanomaterials with excellent adsorption capacities, great chemical reactivity, active atomicity, and environmentally friendly performance have attracted widespread interest as potential adsorbents for heavy metal removal. This review first introduces the application of nanomaterials for removing heavy metal ions from the environment. Then, the environmental factors affecting the adsorption of nanomaterials, their toxicity, and environmental risks are discussed. Finally, the challenges and opportunities of applying nanomaterials in environmental remediation are discussed, which can provide perspectives for future in-depth studies and applications.

**Keywords:** nanomaterials; heavy metals; remediation; nanotoxicity

## 1. Introduction

As a result of extensive industrialization and urbanization over the past century, large amounts of heavy metal ions have been and continue to be discharged into the environment by human activities, such as electroplating, mining, chemical manufacturing, and the application of pesticides and fertilizers [1–3]. Heavy metal contamination in soil and water has become a major problem for many countries throughout the world [4,5]. Because of the non-biodegradable, persistent, and toxic nature of heavy metals, such as Cr(VI), Cd(II), Pb(II), Cu(II), and Hg(II), the ecological environment and human health are seriously threatened [6–8]. For example, the microbial biomass of soils contaminated with Cd, Pb, and Cr is seriously inhibited [9]. Moreover, even low concentrations of heavy metals present in the environment may cause serious environmental and health problems [10,11]. Therefore, in order to protect the ecological environment and public safety, it is imperative to remove these heavy metal ions from contaminated environments.

During the past few decades, numerous treatment methods have been developed to deal with heavy metal contamination, including physical methods, such as adsorption, coagulation, evaporation, and filtration; chemical methods, such as chemical precipitation, oxidation, ion exchange, and electrochemical processes; and biological methods, such as biodegradation and phytoremediation [12–15]. However, most of these treatment methods have significant drawbacks, such as high costs, complexity of operation, and secondary pollution [16–18]. For instance, despite the great removal efficiency of chemical

precipitation, its installation cost is quite high [19]. Of all of the known methods, adsorption is widely used because of its low cost, high removal efficiency, strong practicality, high applicability, and good operability [20,21].

Absorbency is a key factor of the adsorption method. Therefore, it is crucial to select the most suitable adsorption material. A good adsorbent should have the advantages of a large specific surface, great sorption sites, diverse surface interactions, fast adsorption rates, and low costs [22–24]. Currently, the most commonly used adsorbents are biochar, activated carbon, carbon film, biopolymers, clay materials, and nanomaterials [25–27].

Nanomaterials are defined as materials that contain particles measuring between 1.0 and 100 nm in at least one dimension [28,29]. Since the emergence of nanomaterials in the 1970s, an increasing number of researchers have focused on the application of nanomaterials in removing pollutants, such as heavy metals, organic pollutants, and pathogens, from contaminated surface waters, groundwater, sediments, and soil [27,30,31]. Nanomaterials are promising adsorbents and catalysts for the application of environmental remediation because of their great chemical reactivity, large adsorption surface, low temperature modification, and active atomicity [32,33]. The small size of nanoparticles makes it easier for the atoms at the surface to adsorb and have reactions with other atoms in order to achieve charge stabilization [34]. The large specific surface area can greatly improve the adsorption capacities of adsorbents [29,35]. In addition, because of the reduced size, nanomaterials have surfaces that are very reactive [36]. Not only can they efficiently adsorb pollutants, but they also have unique redox properties that are beneficial for the removal of redox-sensitive pollutants via degradation [28,37]. Studies on the removal of heavy metals using nanomaterials are of increasing importance and academic interest, as can be seen from the number of papers published every year, as shown in Figure 1. However, some of the commonly used nanomaterials do have limitations, including high costs, potential toxicity, difficulty in recycling, and an easy interaction with other media [16]. Even though nanomaterials have been widely studied in the field of heavy metal remediation, a comprehensive and systematic review of the application of nanomaterials for the removal of heavy metal ions is relatively lacking.

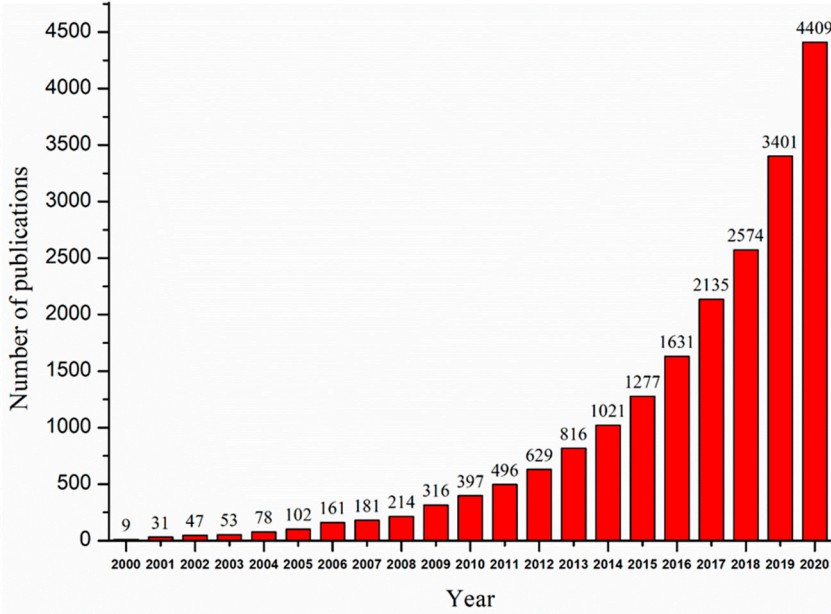

**Figure 1.** Trend of published papers on nanomaterials for removing heavy metals. Obtained from ScienceDirect. Search words (nanomaterials; heavy metal).

Therefore, this paper provides an overview of the application and related research of nanomaterials for removing heavy metal ions from contaminated soil and water and their performance in heavy metal remediation. Additionally, the environmental factors affecting

the adsorption of nanomaterials, their toxicity, and environmental risks after application are discussed in detail. This review provides a reference for future large-scale applications of nanomaterials in remediation projects. Finally, the challenges and opportunities for the researchers who are working hard in this promising field of study are discussed.

## 2. Types of Nanomaterials to Remove Heavy Metals

Nanomaterials are classified into carbon-based nanomaterials and inorganic nanomaterials [38]. They have been widely applied in the field of environmental remediation. Among them, nano zero-valent iron (NZVI), carbon nanotubes (CNTs), and titanium dioxide nanoparticles (TiO$_2$ NPs) are the most frequently used and studied nanomaterials [39,40]. Table 1 summarizes the applications and the performance highlights of nanomaterials for removing heavy metals from water and soil environments.

**Table 1.** Applications of nanomaterials in removing heavy metals from the environment.

| Types of Nanomaterials | Environment | Target Heavy Metals | Performance Highlights | References |
|---|---|---|---|---|
| NZVI-HCS | Water | Pb(II), Cu(II), and Zn(II) | The maximum adsorption capacities were 195.1, 161.9, and 109.7 mg·g$^{-1}$ for Pb(II), Cu(II), and Zn(II), respectively | [41] |
| NZVI | Sediment | Cd(II) | The maximum adsorption capacity of for Cd(II) was 769.2 mg g$^{-1}$ at 297 K | [42] |
| BC-NZVI | Water | Cr(VI) | The performance of BC-NZVI was pH dependent, with a maximum Cr(VI) removal efficiency of 98.71% at pH 2 | [43] |
| BC-NZVI | Soil | Cr(VI) | The immobilization efficiency of Cr(VI) and total Cr reached 100% and 92.9%, respectively, when 8 g kg$^{-1}$ of BC-NZVI was applied for 15 d | [44] |
| NZVI | Water | Pb(II) | The maximum adsorption capacity of NZVI was 807.23mg·g$^{-1}$ at pH 6 | [45] |
| OA-NZVI | Soil | Cd(II), Pb(II), and Zn(II) | The highest Cd, Pb, and Zn removal efficiencies were 46.66%, 48.88% and 47.01%, respectively, for farmland soil at the NZVI concentration of 0.4 g L$^{-1}$ | [46] |
| MWCNTs | Water | Zn(II) | The maximum adsorption efficiency was 96.27% at pH 5 for 6 h | [47] |

**Table 1.** *Cont.*

| Types of Nanomaterials | Environment | Target Heavy Metals | Performance Highlights | References |
|---|---|---|---|---|
| MWCNTs-COOH | Water | Hg(II) and As(III) | The maximum removal efficiencies for Hg(II) and As(III) were 80.5% and 72.4% at the adsorbent dose of 20 mg $L^{-1}$ and pH 7.6–7.9, respectively | [48] |
| CNTs | Water | Zn(II) | The maximum adsorption capacities of Zn(II) were 43.66 and 32.68 mg $g^{-1}$ by SWCNTs and MWCNTs, respectively | [49] |
| TiO$_2$-NCH | Water | Cd(II) and Cu(II) | The maximum adsorption efficiency of Cu(II) and Cd(II) from wastewater samples were 88.01% and 70.67%, respectively | [50] |
| Mesoporous carbonated TiO$_2$ NPs | Water | Sr(II) | The maximum adsorption capacity of Sr(II) 204.4 mg $g^{-1}$ at the natural pH by 4C-TiO$_2$ | [51] |
| TiO$_2$ NPs | Soil | Cd(II) | The greatest Cd accumulation capacity of *Trifolium repens* reached 1235 μg $pot^{-1}$ with PGPR + 500 mg $kg^{-1}$ TiO$_2$ NPs treatment | [52] |

*2.1. Nano Zero-Valent Iron (NZVI)*

NZVI is the most widely studied and applied nanomaterial in environmental remediation and has been proven to be an effective adsorbent, reductant, and catalyst for a variety of contaminants, such as heavy metal ions, halogenated organic compounds, organic dyes, and pharmaceuticals [46–48]. NZVI has a typical core shell structure generated during the synthesis process that contains a shell of Fe(II), Fe(III), and zero-valent iron [53]. As a result of the unique structure, NZVI has the abilities of reduction, surface sorption, stabilization, and precipitation of various contaminants [54–56]. Several studies have reported that NZVI exhibited excellent performance for removing heavy metal(loid) ions from contaminated environments [41–44]. For instance, Yang et al. [41] applied a corn stalk-derived, biochar-supported NZVI for the removal of heavy metal ions from water. The results showed that the equilibrium adsorption capacities reached 195.1, 161.9, and 109.7 mg·$g^{-1}$ for Pb(II), Cu(II), and Zn(II) after 6 h, respectively. Boparai et al. [42] reported that NZVI could be applied as an efficient adsorbent to remove cadmium from contaminated water. The maximum adsorption capacity of NZVI for Cd(II) was 769.2 mg $g^{-1}$, which was achieved at a temperature of 297 K. Su et al. [44] found that the immobilization efficiency of Cr(VI) reached 100% when 8 g $kg^{-1}$ of biochar-supported NZVI was applied in hexavalent chromium-contaminated soil for 15 days. Acid mine water was treated using NZVI, and this resulted in a significant decrease in the concentrations of microcontaminants, such as U, V, As, Cr, Cu, Cd, Ni, and Zn [57]. Huang et al. [58] investigated the effects of

different dosages of NZVI on plant growth and the Pb accumulation of *Lolium perenne*. The Pb accumulation and plant biomass were significantly enhanced when the NZVI and Pb accumulation in *L. perenne* reached a maximum of 1175.40μg per pot with the treatment of 100 mg kg$^{-1}$ NZVI. Vítková et al. [59] reported that NZVI application significantly stabilized the As and Zn in As-rich and Zn-rich soils by the formation of Fe (hydr)oxides. Han et al. [60] investigated the removal efficiency of permeable reactive barriers (PRBs) filled with zero-valent iron (ZVI) and zero-valent aluminum (ZVAl) as a reactive medium and discussed the reaction mechanism of Cr(VI), Cd(II), Ni(II), Cu(II), and Zn(II) with ZVI/ZVAl. The main possible mechanisms were adsorption, formation of metal hydroxide precipitates, and reduction, which are shown in Figure 2.

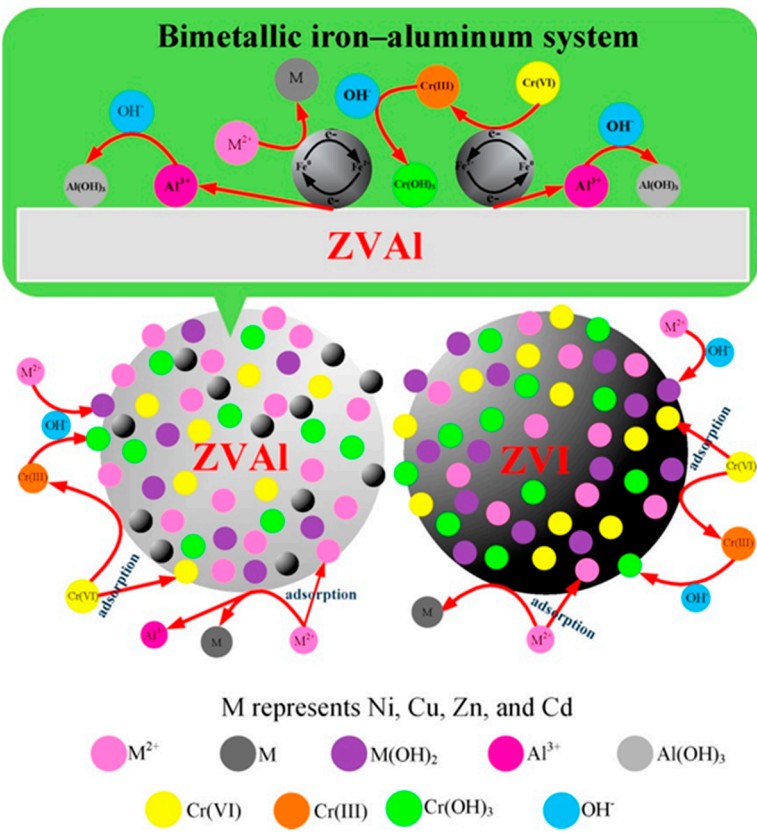

**Figure 2.** Removal mechanisms of five heavy metal ions by zero-valent iron/zero-valent aluminum (ZVI/ZVAl). Reproduced with permission from ref 60 published by Elsevier, 2016.

### 2.2. Carbon Nanotubes (CNTs)

CNTs, which were first discovered in 1991, have a unique chemical structure that consists of a graphitic sheet rolled up in a cylindrical shape [61,62]. CNTs are very strong materials that are over 100 times more resistant and six times lighter than steel [63]. Depending on the number of cylindrical shells, CNTs are classified into two categories: single wall CNTs (SWCNTs) and multi-wall CNTs (MWCNTs). Because of their extraordinary characteristics, such as a large specific surface area, unique morphology, and high reactivity, CNTs are considered to be an excellent nanomaterial for the removal of various organic and inorganic pollutants [64,65]. CNTs can be produced via methods such as chemical vapor deposition, laser ablation, and arc discharge. The adsorption capacity of CNTs is greatly affected by the methods by which they are synthesized with different reactants and catalysts [66]. For instance, Mubarak et al. [67] studied the effect of microwave-assisted MWCNTs on the removal of Zn(II) from wastewater. The results showed that the highest removal rate (99.9%) was achieved at pH 10 and a CNTs dosage of 0.05 g. Sun et al. [68] found that the removal efficiency of Cd(II) by CNTs increased at pH 3. Osman et al. [69]

reported that CNTs synthesized from potato peel-waste material removed up to 84% of Pb(II) within 1 h of the CNTs' application. Yaghmaeian et al. [70] used MWCNTs as a sorbent to remove Hg(II) from wastewater. The results showed that an adsorption capacity of 25.64 mg g$^{-1}$ and a removal rate of greater than 85% were achieved when operated at 25 °C, pH 7, with a contact time of 120 min. Sobhanardakani et al. [71] prepared oxidized MWCNTs and used it as an adsorbent for the removal of Cu(II) from an aqueous solution. The maximum removal rate for Cu(II) was 99.5% at the optimum temperature (25 °C) and the most suitable pH value (6.0). There may be various pathways for heavy metal removal by CNTs, including adsorption, electrostatic interaction, reduction, and ion exchange, depending on the novel properties provided by functionalization and the heavy metal ions (Figure 3).

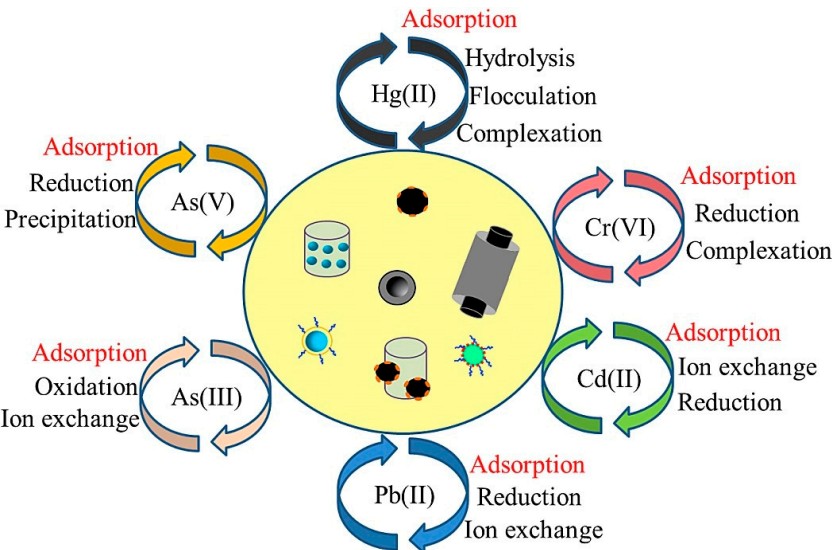

**Figure 3.** The mechanisms of removing heavy metal from an aqueous environment by carbon nanotubes (CNTs). Reproduced with permission from ref [72] published by Elsevier, 2018.

### 2.3. Titanium Dioxide Nanoparticles (TiO$_2$ NPs)

Among the nanomaterials used for environmental remediation, TiO$_2$ NPs have been extensively studied [73]. TiO$_2$ NPs show good abilities for photocatalysis, high reactivity, and chemical stability, and they have been successfully applied for modifying the mobility and toxicity of heavy metals in water, soil, and sediment [74,75]. In addition, another advantage of TiO$_2$ NPs is their ease of synthesis. Goutam et al. [76] synthesized TiO$_2$ NPs using a leaf extract and used it to treat tannery wastewater. The results showed that 76.48% of the Cr was removed from the wastewater using green-synthesized TiO$_2$ NPs. Mahmoud et al. [77] reported that the microwave-synthesized TiO$_2$ NPs bonded with the chitosan nanolayer and removed 88.01% of the Cu (II) and 70.67% of the Cd (II) from wastewater when the pH value was 7.0. Gebru et al. [78] synthesized cellulose acetate (CA)/TiO$_2$ NPs using a new electrospinning technique and tested its adsorption capacity for removing Pb(II) and Cu(II) ions from water. The CA/TiO$_2$ adsorbent removed 99.7% and 98.9% of Pb(II) and Cu(II) ions under the most optimized conditions. Fan et al. [79] reported that the concentrations of exchangeable, carbonate, and iron-manganese oxide of As and Pb in the sediments decreased with an increasing amount of TiO$_2$ NPs. Singh and Lee [80] investigated the effect of TiO$_2$ NPs on Cd phytoremediation in *Glycine max*. The results showed that the Cd accumulation in the aerial portions of the plants increased by approximately 2.6 times when 300 mg kg$^{-1}$ TiO$_2$ NPs were added to the soil. Zhao et al. [81] proposed the possible removal mechanisms of Cr(VI) by reduced graphene oxide decorated with TiO$_2$ NPs (TiO$_2$-RGO), which is shown in Figure 4. It was speculated that the negatively charged Cr(VI) was bound to the surface of TiO$_2$-RGO, which had a positive

charge and was reduced to Cr(III). Then, the Cr(III) species was released into the solution due to electrostatic repulsion with the surface of $TiO_2$-RGO.

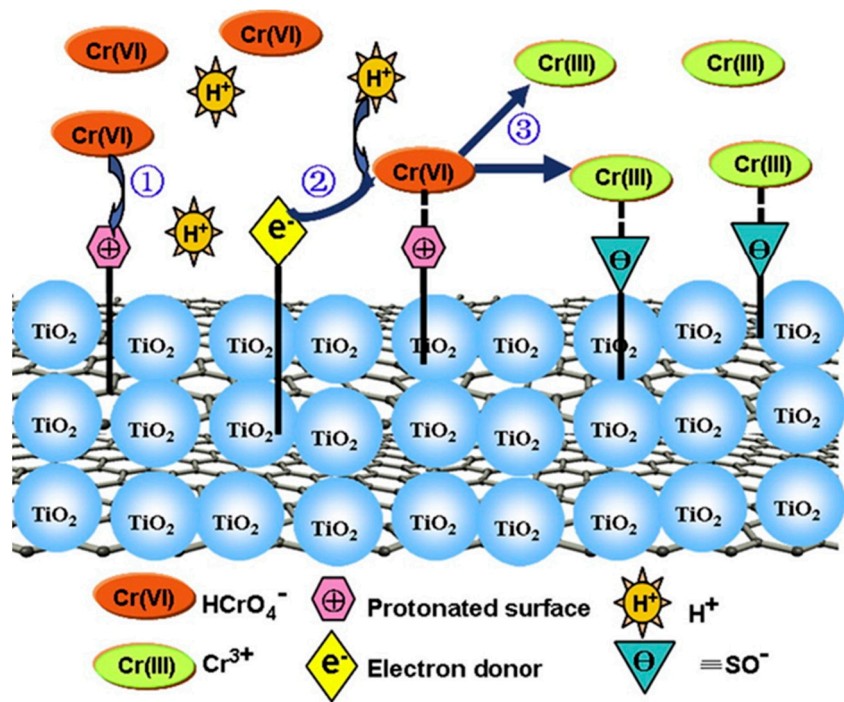

**Figure 4.** The possible mechanism of Cr(VI) reduction by $TiO_2$-RGO. Reproduced with permission from ref 81 published by Elsevier, 2013.

## 3. Environmental Factors Affecting the Performance of Nanomaterials

### 3.1. The Effect of pH

The solution pH is an important parameter in the reactions of nanomaterials with heavy metal ions [25]. The solution pH value affects the surface charge of the nanomaterial and thus affects its adsorption ability. In addition, the pH determines the bioavailability and existent forms of heavy metal ions [82]. At a lower pH, more protons react with nanomaterials, such as NZVI, and the conversion of $H^+$ to $H_2$ can result in more reactive atomic hydrogen and a faster reduction rate. The surface coordination, electrostatic sorption, and precipitation become stronger under neutral pH conditions, which leads to a higher removal rate [83]. For example, Zhao et al. [84] reported that the adsorption ability of NZVI decreased significantly under strong alkaline or acidic conditions. The NZVI corrosion was affected by the pH, and thus affected its reactive lifetime. Liu et al. [85] investigated the effects of the pH on the removal efficiency of Hg(II) and Cr(VI) by a pumice-supported NZVI from an aqueous solution. The results showed that when the pH increased from 3.1 to 8.1, the Hg(II) removal rates increased, while the Cr(VI) removal rates decreased. Wu et al. [86] found that the Cr(VI) removal rate of FeS nanoparticles stabilized by sodium alginate increased from 92.51% to 99.65% when the pH was increased from 4.0 to 6.0, while the Cr(VI) removal rate decreased to 65.37 when the pH was 10.0. Xu and Zhao [87] investigated the effect of the pH on the Cr(VI) immobilization in contaminated soil using carboxymethyl cellulose-stabilized NZVI. The results showed that the Cr(VI) leached from the soil reduced from around 30% to 20%, with the soil pH decreasing from 9.0 to 5.0.

### 3.2. The Effect of the Contact Time

Generally, the contact time between nanoparticles and heavy metal ions can significantly affect the removal rates during the adsorption and redox process [88]. Several studies have investigated the effects of contact time by applying models such as the pseudo-first-order, the pseudo-second-order, the Zeldowitsch, and the Lagergren kinetic models [89].

Typically, the adsorption rate of heavy metal ions onto nanomaterials quickly reached the highest point in the beginning phase and then slowed down with time until the sorption equilibrium was reached. Specifically, Gong et al. [90] found that the Hg(II) removal rate by the sodium carboxymethyl cellulose-stabilized FeS nanoparticles reached the highest point within 30 minutes and slowly decreased until equilibrium was achieved at approximately 6 h. It was reported by Lv et al. [91] that Cr(VI) was removed rapidly by NZVI–$Fe_3O_4$ nanocomposites within 2 h and then slowed down until equilibrium. The kinetics model was described well by the pseudo-second-order model. Cao et al. [46] investigated the removal efficiency of Cd(II), Pb(II), and Zn(II) from mine- and farmland-contaminated soils using the soil-washing method with the application of NZVI combined with low-molecular-weight organic acids. The results showed that the removal efficiency of heavy metals increased rapidly in the first 2 h and then slowed down until equilibrium. The initial process of 2 h was described by the pseudo-first-order model, and the whole process of 12 h was described by the pseudo-second-order model.

### 3.3. The Effect of the Adsorbent Dosage

The dosage of the nanomaterials used is another key factor that affects the removal capacities of heavy metal ions. Many publications have selected the optimum dosage of adsorbents that can achieve the desired removal efficiency, which is useful for the cost-effective application of nanomaterials. Arshadi et al. [92] reported that the Pb(II) removal rate by immobilized NZVI on the sineguelas waste biomaterial increased from 15.6% to 89% when the adsorbent dosage increased from 0.05 to 0.15 g. However, a higher adsorbent dosage did not result in a significant increase in the removal rate of 89%. Fu et al. [32] investigated the removal efficiency of Cr(VI) and Pb(II) from groundwater by sepiolite-supported NZVI. The results showed that when the adsorbent dosage was increased from 0.05 to 3.2 g $L^{-1}$, the Cr(VI) removal rate was raised from 45.1% to 99.2%, and the Pb(II) removal rate was raised from 56.2% to 99.9%. However, a dosage of 1.6 g $L^{-1}$ sepiolite-supported NZVI was selected as the optimal dosage because the pseudofirst-order rate constants of Cr(VI) and Pb(II) did not increase significantly after the adsorbent dosage of 1.6 g $L^{-1}$. Zand et al. [52] investigated the phytoremediation of the Cd contaminated soil with the application of different doses of $TiO_2$ NPs. The results showed that the Cd uptake by *Trifolium repens* was significantly enhanced when the application dosage of $TiO_2$ NPs was increased from 0 to 500 mg $kg^{-1}$. The application of 1000 mg $kg^{-1}$ $TiO_2$ NPs resulted in a significant reduction of plant biomass due to toxic effects.

### 3.4. The Effect of Temperature

The temperature determines the energy of reaction activity and thus plays key roles in the adsorption process. The increase or decrease in temperature can alter the equilibrium adsorption capacity of nanomaterials. Furthermore, the higher temperature can reduce the distance between nanoparticles and increase the redox reaction rate. Dubey et al. [93] studied the removal efficiency of Hg(II) by chitosan–alginate nanoparticles when the temperature ranged from 10 to 40 °C. The results showed that the removal efficiency increased with the increasing temperature until 30 °C and then started to decrease. Similar results were also reported elsewhere [94]. Nassar [95] reported that the Pb(II) adsorption by $Fe_3O_4$ nanomaterials increased with the increase of temperature from 298 to 328 K, which indicated that the adsorption process was endothermic. Liu et al. [96] investigated the immobilization efficiency of Re(VII) using starch-stabilized NZVI in soil and groundwater. The results showed that the immobilization efficiency of Re(VII) increased with the increasing temperature from 15 to 45 °C. The results can be explained by the classical Arrhenius equation.

### 4. Environmental Impacts of Nanomaterials

Nanomaterials have provided a wide range of applications for reducing/immobilizing metal(loid)s in contaminated water and soil [96,97]. However, the massive use of nano-

materials will inevitably result in their elevated concentrations in the environment, which may affect ecological security and human health [98–104]. The toxicity of nanomaterials and their ability to change the bioavailability of toxic contaminants such as heavy metals should not be neglected [105–107]. It is of great importance to investigate the environmental impacts of nanomaterials due to their increasing use in the remediation of contaminated water and soil. Lu et al. [106] reported that the toxicity of Cd(II) for *Artemia salina*, a model marine zooplankton, increased by 12.2% when 5 mg $L^{-1}$ of TiO$_2$ NPs was added. However, when the concentration of TiO$_2$ NPs was increased to 400 mg $L^{-1}$, the toxicity of Cd(II) was reduced to 57.1%, which indicated a concentration-dependent toxicity of nanomaterials. Deng et al. [107] investigated the physiological and biochemical responses of *Phaeodactylum tricornutum* to TiO$_2$ NPs. The results showed that the growth inhibition rate of *P. tricornutum* increased from 5.46% to 27.31% when the dosage of TiO$_2$ NPs increased from 2.5 to 40 mg $L^{-1}$ at an exposure of 96 h. Lam et al. [108] found that the mice treated with high dosages of single-wall carbon nanotubes revealed peribronchial inflammation, while the mice treated with carbon black were normal. Lindberg et al. [109] revealed that CNTs induced a dose-dependent increase in DNA damage assessed by a single cell gel electrophoresis assay and a micronucleus assay in human bronchial epithelial BEAS 2B cells. A study showed that a dosage of NZVI up to 10 mg $L^{-1}$ resulted in a doubling of the decrease in fertilization success of marine organisms including mussels, sea squirts, and urchins [110]. El-Temsah and Joner [111] reported that NZVI had toxicity effects on soil microorganisms (ostracods and collembola), especially after a 7-d incubation. However, the toxicity effect was observed to be alleviated with the increase in incubation time. Fajardo et al. [112] investigated the impacts of NZVI on soil microbial structures and functionality. The results showed that the application of 10 mg $mL^{-1}$ NZVI had no significant effect on the cellular viability and biological activity of the soil microorganisms. The FISH assays showed that NZVI promoted the dominance of some microbial groups and thus changed the soil microbial structure. As discussed in Section 3.3, the dosage of nanomaterials is a key factor for removing heavy metals from the environment. However, the toxicity of nanomaterials has also been shown to be dosage dependent [113]. In conclusion, previous studies showed that nanomaterials had toxicity effects on microorganisms, aquatic organisms, and plants [106–113]. The feasibility of applying nanomaterials in the remediation of contaminated water and soil should be questioned. However, it is worth noticing that the toxicity of nanomaterials is affected by the tested organism species, the dosage of nanomaterials, and the environmental factors. Therefore, it is difficult to conduct a comprehensive and in-depth analysis of the toxicity effects of nanomaterials. Moreover, the selection of a suitable modification method, synthesis method, and the dosage of nanomaterials can minimize the adverse effect on the environment.

## 5. Conclusions and Future Perspectives

Nanomaterials are revolutionary materials with properties that include nanoscale size, large specific surface area, and great reactivity. According to the current knowledge, nanomaterials have substantial potential for remediating heavy metal-contaminated water and soil. In this review, the applications, environmental factors, toxicity, and future perspectives of nanomaterials on heavy metal remediation were discussed, as shown in Figure 5. Their applications in the field of environmental pollution control, especially in heavy metal remediation will certainly continue to be studied. Despite the many advantages of nanomaterials, there are still several challenges in their application for heavy metal remediation that require attention in future studies, including: (1) the toxicity of nanomaterials on the plants, animals, and the microbial community; (2) the management and regeneration of nanomaterials; (3) the recovery of heavy metal ions from saturated nanomaterials; (4) the methods to decrease the aging of nanomaterials; (5) the combined application of nanomaterials with other treatments such as phytoremediation; (6) the synergistic or antagonistic effects of nanomaterials and microbial activities; and (7) the long-term stability of heavy metal remediation by nanomaterials, especially in field studies.

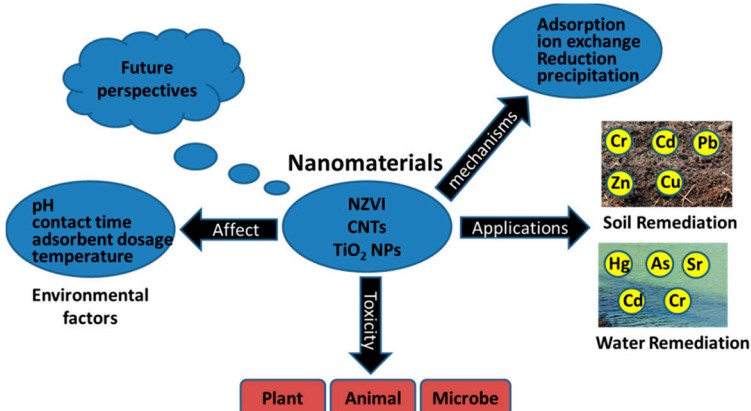

**Figure 5.** Applications of nanomaterials for heavy metal removal from water and soil.

**Author Contributions:** Conceptualization, G.Y. and P.J.; methodology, X.W.; software, N.D.; validation, S.Y., Q.G. and F.L.; formal analysis, G.Y.; investigation, P.J.; resources, G.Y.; data curation, X.W.; writing—original draft preparation, G.Y.; writing—review and editing, J.L.; visualization, P.J.; supervision, J.L.; project administration, J.L.; funding acquisition, J.L. All authors have read and agreed to the published version of the manuscript.

**Funding:** This research was sponsored by the National Science Foundation of China (Grant No. 51868010, 41867022), the Special Funds of Guangxi Distinguished Experts, the Program for High Level Innovation Team and Outstanding Scholar of Universities in Guangxi (Grant No. GuiCaiJiaoHan [2018]319), the Guangxi Science and Technology Project (2018AD16013-04), and the Guilin Science and Technology Project (Grant No. 20190219-3).

**Institutional Review Board Statement:** Ethical review and approval were waived for this study.

**Informed Consent Statement:** Not applicable.

**Data Availability Statement:** All data, models, and code generated or used during the study appear in the submitted article.

**Acknowledgments:** The authors wish to thank the anonymous reviewers and the editor for their comments and suggestions.

**Conflicts of Interest:** The authors declare no conflict of interest.

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
