# Peer review of "Applications of Nanomaterials for Heavy Metal Removal from Water and Soil: A Review"

_sustainability, doi:10.3390/su13020713_

Round 1
Reviewer 1 Report
SUMMARY OF THE REVIEW:
The submitted research is reviewing a very interesting issue related to the remediation of heavy metal contaminated water and soil using nanomaterials. I found this paper very interesting where Several technical aspects were nicely implemented and explained sufficiently. Undoubtedly, authors invested huge amount of time and have made a great effort to produce this high-quality of research which is clearly structured and the language used is largely appropriate. I see that this manuscript in its form and level deserves to be accepted for publication in MDPI-sustainability but after considering below MINOR COMMENTS.
DETAILED COMMENTS:
The title: The title of this review paper is perfect.
ABSTRACT: I found the abstract very well structured and contains the necessary information.
1.INTRODUCTION: The introduction is well written and very organised. I suggest for the authors to mention references showing the techniques of quantifying heavy metals in water and soil, and I suggest the following: https://doi.org/10.3390/s19040762 and https://doi.org/10.1016/j.jhazmat.2011.04.090
2.Types of nanomaterials to remove heavy metals: This section of the paper has the required information and the authors successfully explained the mechanisms of nanoparticles adsorption.
3.Factors affecting the performance of nanomaterials: The authors provided three factors that affect the performance of the nanomaterials, good and high-level information have been discussed under this section.
4.Environmental impacts of nanomaterials: I appreciate adding this section about the environment, authors highlighted successfully the issue toxicity/dosage of nanomaterials.
5.Conclusions and future perspectives: The authors closed with a good conclusion.
As final general comment, please make sure to define ALL the acronyms form their first appearance in the paper. Also, all the references MUST BE CHECKED and formatted as required by MDPI-Sustainability, also make sure that all the references have DOI number unless it is not available.
Author Response
Response to Reviewer 1 Comments
We are truly grateful to the critical comments and thoughtful suggestions. Based on these comments and suggestions, we have made careful modifications on the original manuscript. All changes made to the text are indicated using the track changes mode. Our point-by-point responses are listed below.
Point 1: I suggest for the authors to mention references showing the techniques of quantifying heavy metals in water and soil, and I suggest the following: https://doi.org/10.3390/s19040762 and https://doi.org/10.1016/j.jhazmat.2011.04.090
Response 1: The suggested references have been mentioned in the revised manuscript. The reference numbers were 95 and 96.
Point 2: As final general comment, please make sure to define ALL the acronyms form their first appearance in the paper. Also, all the references MUST BE CHECKED and formatted as required by MDPI-Sustainability, also make sure that all the references have DOI number unless it is not available.
Response 2: We have checked and made sure that all the acronyms were defined form their first appearance in the manuscript. The formats of all the references were checked. We have added all the available DOI numbers of the references.
Reviewer 2 Report
Here is my report about the MS: Applications of nanomaterials for heavy metal removal from water and soil : A review; that was submitted by Yu et al. The MS reviewed the remediation of heavy metal contaminated water and soil using nanomaterials, since this is an important issue for the environmental and biological point of views.
The English of the MS is poor in some places.
L98-99: Several studies have….. Where ?? I do not see them! Authors must cite them.
Environmental impacts of nanomaterials are not well presented, authors should deeply write this part.
Since there is a negative impact for nanomaterials on plants, animals, microorganisms, why we use it?
I was looking to see your own Figures as a conclusion of your background on this topic, instead of using the already published Figures.
I really recommend the authors to work hard on their MS and improved it, so it will be suitable for sustainability.
The following recent and related references are highly recommended to be cite in this MS:
- Seleiman, M.F.; Almutairi, K.F.; Alotaibi, M.; Shami, A.; Alhammad, B.A.; Battaglia, M.L. Nano-Fertilization as an Emerging Fertilization Technique: Why Can Modern Agriculture Benefit from Its Use? Plants 2021, 10, 2.
- Seleiman, M.F.; Alotaibi, M.A.; Alhammad, B.A.; Alharbi, B.M.; Refay, Y.; Badawy, S.A. Effects of ZnO Nanoparticles and Biochar of Rice Straw and Cow Manure on Characteristics of Contaminated Soil and Sunflower Productivity, Oil Quality, and Heavy Metals Uptake. Agronomy 2020, 10, 790.
- Seleiman, M.F., Ali, S., Refay, Y., Alhammad, B.A., El-Hendawy, S.E. 2020. Chromium resistant microbes and melatonin reduced Cr uptake and toxicity, improved physio-biochemical traits and yield of wheat in contaminated soil. Chemosphere, 2020, 250, 126239
Author Response
Response to Reviewer 2 Comments
We are truly grateful to the critical comments and thoughtful suggestions. Based on these comments and suggestions, we have made careful modifications on the original manuscript. All changes made to the text are indicated using the track changes mode. Our point-by-point responses are listed below.
Point 1: L98-99: Several studies have….. Where ?? I do not see them! Authors must cite them.
Response 1: This sentence has been cited.
Point 2: Environmental impacts of nanomaterials are not well presented, authors should deeply write this part. Since there is a negative impact for nanomaterials on plants, animals, microorganisms, why we use it?
Response 2: The part “Environmental impacts of nanomaterials” has been rewritten. More contents were added to support the point of view.We emphasized that the the environmental impacts of nanomaterials on microorganism, plants and aquatic organism are affected by many factors. Besides, the selection of suitable modification method, synthesis method, and dosage of nanomaterials can minimize the adverse effect on the environment.
Point 3: I was looking to see your own Figures as a conclusion of your background on this topic, instead of using the already published Figures.
Response 3: Thanks for the nice advice. A graphical abstract was made to summarize the main points of this review.
Point 4: The following recent and related references are highly recommended to be cite in this MS:
Seleiman, M.F.; Almutairi, K.F.; Alotaibi, M.; Shami, A.; Alhammad, B.A.; Battaglia, M.L. Nano-Fertilization as an Emerging Fertilization Technique: Why Can Modern Agriculture Benefit from Its Use? Plants 2021, 10, 2.
Seleiman, M.F.; Alotaibi, M.A.; Alhammad, B.A.; Alharbi, B.M.; Refay, Y.; Badawy, S.A. Effects of ZnO Nanoparticles and Biochar of Rice Straw and Cow Manure on Characteristics of Contaminated Soil and Sunflower Productivity, Oil Quality, and Heavy Metals Uptake. Agronomy 2020, 10, 790.
Seleiman, M.F., Ali, S., Refay, Y., Alhammad, B.A., El-Hendawy, S.E. 2020. Chromium resistant microbes and melatonin reduced Cr uptake and toxicity, improved physio-biochemical traits and yield of wheat in contaminated soil. Chemosphere, 2020, 250, 126239
Response 4: The suggested references have been mentioned in the revised manuscript. The reference numbers were 97, 98 and 99.
Reviewer 3 Report
Dear Authors,
Please pay attention to the journal's guidelines for article formatting (the entire manuscript).
Please take a closer look at the PRISMA format and use the checklist to correct the article: http://prisma-statement.org/PRISMAStatement/Checklist.
I also ask you to take care of the quality of the graphics / figures, because the current graph and drawings are unacceptable.
Yours faithfully,
Author Response
Response to Reviewer 3 Comments
We are truly grateful to the critical comments and thoughtful suggestions. Based on these comments and suggestions, we have made careful modifications on the original manuscript. All changes made to the text are indicated using the track changes mode. Our point-by-point responses are listed below.
Point 1: Please pay attention to the journal's guidelines for article formatting (the entire manuscript).
Response 1: We have checked the formatting according to the journal's guidelines.
Point 2: Please take a closer look at the PRISMA format and use the checklist to correct the article: http://prisma-statement.org/PRISMAStatement/Checklist.
Response 2: We have checked the manuscript according to the PRISMA format.
Point 3: I also ask you to take care of the quality of the graphics / figures, because the current graph and drawings are unacceptable.
Response 3: We have adjusted the graphics and figures. Besides, original files of the figures are available and can be sent to the editors via email.
Round 2
Reviewer 2 Report
The ms looks better now.
Reviewer 3 Report
Dear Authors,
thank you for improving the manuscript.
Yours faithfully,
This manuscript is a resubmission of an earlier submission. The following is a list of the peer review reports and author responses from that submission.